# Comparison of Resin Cement's Different Thicknesses and Poisson's Ratios on the Stress Distribution of Class II Amalgam Restoration Using Finite Element Analysis

**Hakan Yasin Gönder** [1,*], **Yasemin Derya Fidancıoğlu** [2], **Muhammet Fidan** [3], **Reza Mohammadi** [4] and **Said Karabekiroğlu** [1]

1   Department of Restorative Dentistry, Faculty of Dentistry, Necmettin Erbakan University, Konya 42090, Turkey
2   Department of Pediatric Dentistry, Faculty of Dentistry, Necmettin Erbakan University, Konya 42090, Turkey
3   Department of Restorative Dentistry, Faculty of Dentistry, Usak University, Usak 64200, Turkey
4   Faculty of Dentistry, Necmettin Erbakan University, Konya 42090, Turkey
*   Correspondence: hygonder@erbakan.edu.tr; Tel.: +90-530-581-55-89; Fax: +90-332-220-00-45

**Abstract:** Using a three-dimensional finite element analysis, this study aimed to evaluate the effect of different cements' thicknesses and Poisson's ratios on the stress distribution in enamel, dentin, restoration, and resin cement in a computer-aided design of a class II disto-occlusal cavity. Dental tomography was used to scan the maxillary first molar, creating a three-dimensional tooth model. A cavity was created with a 95 degree cavity edge angle. Resin cement with varying Poisson's ratios (V1: 0.35 and V2: 0.24) was used under the amalgam. The simulated groups' thicknesses ranged from 50 μm to 150 μm. A load of 600 N was applied to the chewing area. The finite element method was used to assess the stress distribution in the enamel, dentin, restorations, and resin cement. The stress in the restoration increased with the use of a 100 μm resin cement thickness and decreased with the use of a 150 μm resin cement thickness. For the V1 and V2 groups, the cement thickness with the maximum stress value for the enamel and dentin was 150 μm, while the cement thickness with the lowest stress value was 50 μm. The greatest stress values for V1 and V2 were obtained at a 150 μm cement thickness, while the lowest stress values were observed at a 100 μm cement thickness. Using resin cement with a low Poisson's ratio under amalgam may reduce stress on enamel and restorations.

**Keywords:** dental amalgam; cement thickness; finite element method; resin cement; stress distribution

## 1. Introduction

Functional and parafunctional forces can lead to significant stress accumulation in various directions and magnitudes within healthy teeth and dental hard tissues, both before and after dental treatment [1]. Analyzing the distribution of these stresses and determining their magnitudes can be instrumental in reducing the likelihood of restoration failures that may occur in dental restorations [2]. Preserving healthy enamel and dentin tissues is crucial after the cavity preparation and restoration of decayed teeth [3]. While there is currently no restorative material that can match the mechanical and biological properties of natural teeth [4], restorative materials must be capable of replacing both enamel and dentin and exhibit elastic properties that are similar to those of dental tissues [5]. The success of posterior restorations is influenced by several factors, including the type, shape, and size of the cavity; the restoration materials used; and patient and dentist factors. Specifically, class II cavities can significantly weaken the strength of teeth against forces [3].

The use of dental amalgam as a restorative material has a long history, and while it is not the only filling material available, it does offer certain advantages over other materials [6]. One such advantage is its affordability as a direct restorative material, although more expensive resin-based alternatives have been developed [7]. Additionally, dental amalgam

is easier to place than tooth-colored filling materials, such as composites, because it does not require a perfectly isolated working area, which can be challenging in various settings, such as congested hospitals, or for patients with special requirements. Thus, dental amalgam is still frequently used in such institutions due to its ease of use [8]. The use of dental amalgam in restorative applications has been prevalent due to several advantages, including its low cost, easy application, and bacteriostatic properties [6]. However, its popularity is declining due to its poor esthetics and adverse environmental effects [9]. Furthermore, dental amalgam has several disadvantages, such as a poor marginal compatibility, an unesthetic color, the conduction of heat and electricity, and tooth discoloration caused by corrosion [10]. A significant limitation of dental amalgam is its inability to effectively adhere to residual dental tissues, resulting in an inadequate sealing of the restoration to the tooth [11]. Resins offer potential benefits, as they can seal the edges of the restoration between the cement and the tooth structure, providing additional retention to the compromised tooth structure. To minimize material loss in the tooth, an adhesive system is used with a resin cement. Resin cements ensure chemical and mechanical bonding, increasing the adhesion of the amalgam to the tooth structure [12]. Resin cements, with their high compressive and tensile strengths and high bonding value to the tooth and porcelain surface, have the lowest solubility compared to other cements [13]. Using resin cements in restorative procedures can decrease microleakage, increase restorative strength, and enhance tooth longevity [14].

The finite element analysis is one of the most effective analyses used in stress, strength, fluid, vibration, and dynamic calculations. Additionally, it can be defined as a solution method in which complex problems are divided into simple sub-problems, with each one being solved separately [8]. Various external factors, such as occlusal loading, have been used, and studies have been conducted to evaluate the forces acting on teeth and restorations [15]. Stress levels can be assessed with numerical analyses involving the finite element analysis (FEA), bioengineering, and dentistry. Numerical computations are made with this method by loading complex structures with various parameters [9], and stress levels are evaluated [16]. The FEA method separates the region to be analyzed into basic and simple elements that are developed from single parts to a whole. The deformation of the entire structure at each node, the stresses, and the consequent variables may be computed based on the state of the components connected by the nodes [17]. This method is used to design and optimize modern materials utilized in dental reconstructions (crowns and root restorations) and to assess the risk of failed dental treatment occurring as a result of the dental tissue structure or restoration material [16]. This analysis method can be measured in vivo by determining the tooth structure and calculating the stress and strain in biomaterials [9]. Furthermore, in terms of time and cost, the use of numerical models and in vitro simulations is beneficial in laboratory and clinical research [18].

Understanding the stress distribution at the tooth interface during occlusal loading may help to relieve the observed clinical issues and increase the success rate of restorations. The forces acting on tooth restorations or the tooth–restoration interface can be shown among the factors determining the success or failure of dental treatments. It is questioned whether the stress behavior of resin cement affects the longevity of the treatment in teeth with amalgam restorations. Using a three-dimensional finite element analysis, this study aimed to investigate the effect of different cements' thicknesses and Poisson's ratios on the stress distribution in the enamel, dentin, restoration, and resin cement of a class II disto-occlusal cavity.

## 2. Materials and Methods

A flowchart of the finite element analysis is shown in Figure 1. The finite element analysis process began with the use of a dental tomography instrument to obtain a 3D image of the left permanent maxillary first molar tooth (J Morita MFG Corp., Kyoto, Japan) in DICOM format. The DICOM files were imported into the Materialize interactive medical image control system (Mimics 12.00, Leuven, Belgium) program, where different masks were created for each dental tissue (the enamel, dentin, and pulp). The density thresholds

were adjusted manually to accurately depict the anatomy of the tooth. Afterward, 3D objects of each mask were created and converted to STL files (Figure 2). The 3D image was then separated into surfaces using the Geomagic Design X 2020.0 program to generate the appropriate arrangements for the finite element analysis.

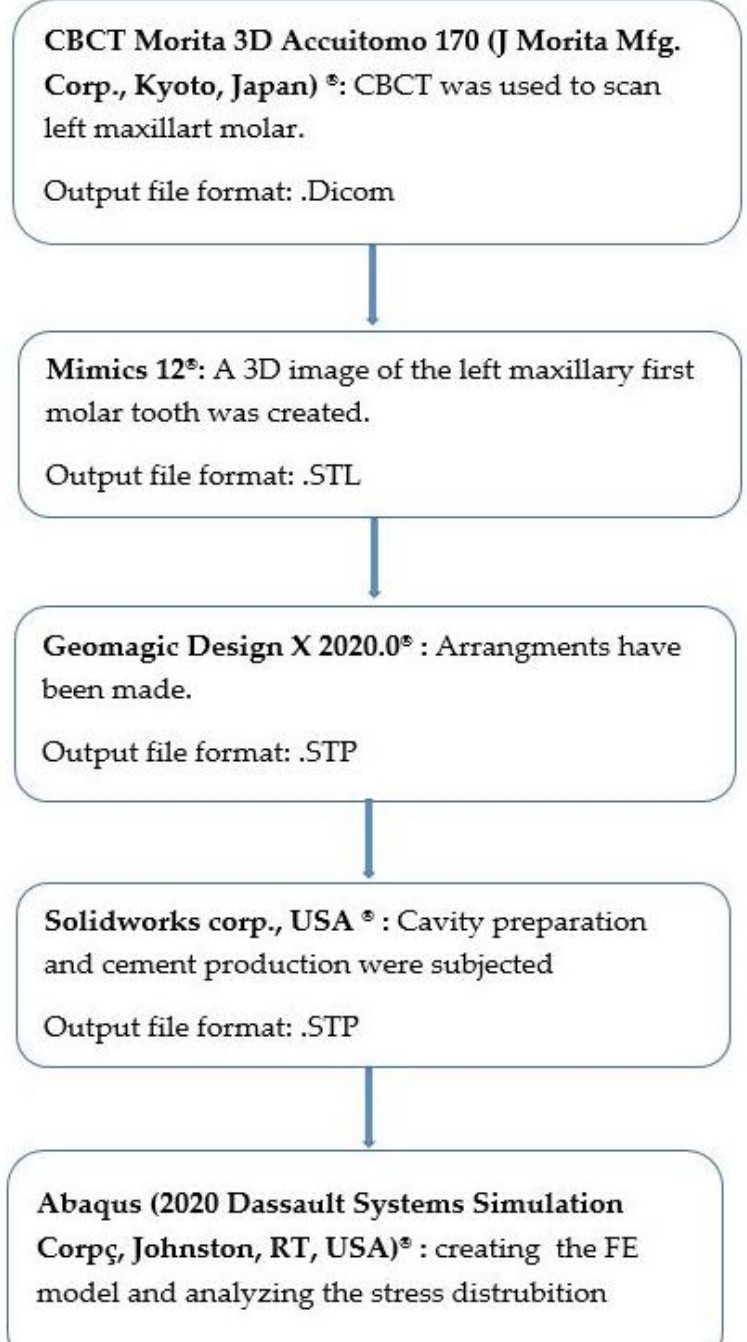

**Figure 1.** Flowchart for performing finite element analysis.

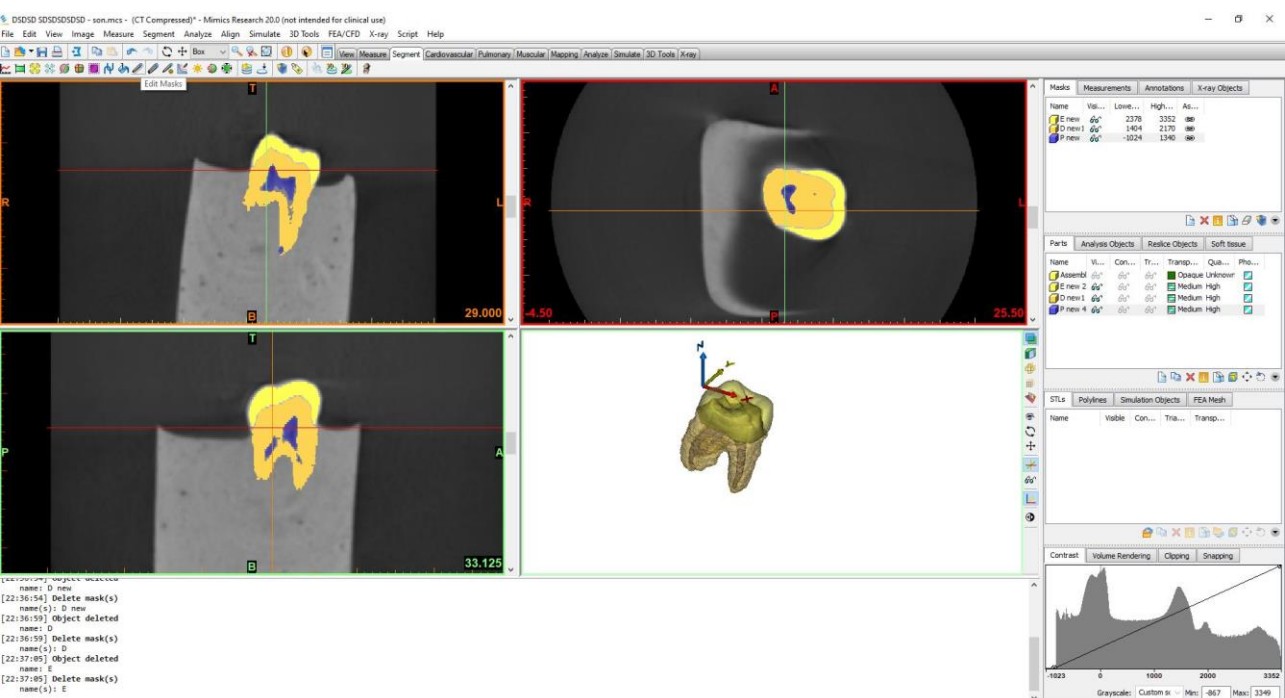

**Figure 2.** CBCT-scan data as seen in MIMICS 12. The tooth is presented in three different cross-sectional views.

The tooth model was placed in a coordinate system, with the *x*-axis indicating the buccolingual direction, the *y*-axis indicating the mesiodistal direction, and the *z*-axis oriented upward. Class II DO cavity modeling was performed on the 3D model using Solidworks 2013 software (Solidworks Corp, Waltham, MA, USA), with a cavity angle of 95 degrees (Figure 3). The class II cavity had an occlusal depth of 4 mm and an occlusal-gingival depth of 6 mm, and it was fixed with the occlusal margin of the enamel and the gingival margin of the dentin. To reduce the stress concentration, the cavity's inner line angles were rounded. The relationship between the elastic modulus, stress, strain, and Poisson's ratio values was explained using Hooke's law, and the values were reported by the programs used in the finite element analysis via this law [19].

$$\varepsilon_{11} = \frac{1}{E}\left[\sigma_{11} - \nu(\sigma_{22} + \sigma_{33})\right]$$

$$\varepsilon_{22} = \frac{1}{E}\left[\sigma_{22} - \nu(\sigma_{11} + \sigma_{33})\right]$$

$$\varepsilon_{33} = \frac{1}{E}\left[\sigma_{33} - \nu(\sigma_{11} + \sigma_{22})\right]$$

$$\varepsilon_{12} = \frac{1}{2G}\sigma_{12}; \; \varepsilon_{13} = \frac{1}{2G}\sigma_{13}; \; \varepsilon_{23} = \frac{1}{2G}\sigma_{23};$$

*σ = Stress; E = Young' modulus; ε = Strain; ν = Poisson's ratio; G = Shear modulus*

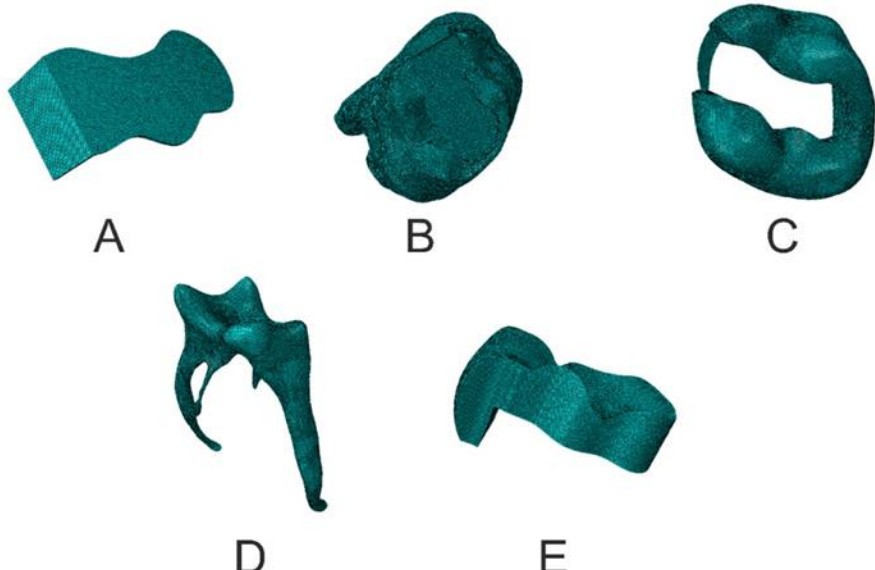

**Figure 3.** Geometric construction of molar tooth, restoration, and cement. (**A**): cement, (**B**): dentin, (**C**): enamel, (**D**): pulp, (**E**): restoration.

The enamel, dentin, resin cement, and material characteristics were assumed to be isotropic and linearly elastic to simplify the complexity of the three-dimensional finite element models. The models were assigned the properties of the teeth and the materials, and the relevant data are presented in Table 1.

**Table 1.** Mechanical properties of the material used in 3D FE models of maxillary molars.

| Material | Young's Modulus (GPa) | Poisson's Ratio | Tensile Strength (MPa) | Compressive Strength (MPa) |
|---|---|---|---|---|
| Dentin | 18.6 [20] | 0.31 [20] | 98.7 [21] | 297.0 [21] |
| Enamel | 84.1 [20] | 0.33 [20] | 10.3 [21] | 384.0 [21] |
| Pulp | 0.002 [22] | 0.45 [22] | - | - |
| Amalgam | 35.0 [7,23] | 0.35 [7,23] | 3–58 [24] | 45–550 [24] |
| Resin Cement (V1) Variolink II (Ivoclar Vivadent, Schaan Liechtenstein) | 8.3 [25] | 0.35 [25] | - | - |
| Resin Cement (V2) Variolink II (Ivoclar Vivadent, Schaan Liechtenstein) | 8.3 [26] | 0.24 [26] | - | - |

Different combinations were simulated, including the same modulus of elasticity for the two cement groups (V1 and V2; Table 1) and different thicknesses (50 μm (A), 100 μm (B), and 150 μm (C)). As a result, six study groups were created (Table 2).

**Table 2.** Study groups.

| Study Group | Cement Thickness |
|---|---|
| V1A | 50 μm |
| V1B | 100 μm |
| V1C | 150 μm |
| V2A | 50 μm |
| V2B | 100 μm |
| V2C | 150 μm |

V1: Variolink; Poisson's ratio: 0.35; V2: Variolink; Poisson's ratio: 0.24; 50 μm: A; 100 μm: B; and 150 μm: C.

As the periodontal ligament (PDL) was not modeled, fixed and pinned boundary conditions were utilized to simulate the roots that are fixed in the bone [8,27]. A single tooth and tooth type were used without simulating the periodontal ligament or bone. The mechanical boundary conditions (symmetry/antisymmetry/encostre) were selected using the "create boundary condition" tab in the load part of the Abaqus program. The effect of the periodontal ligament was ignored, and the tooth was pinned (U1 = U2 = U3 = 0) from the enamel–cementum junction to the apical region (Figure 4).

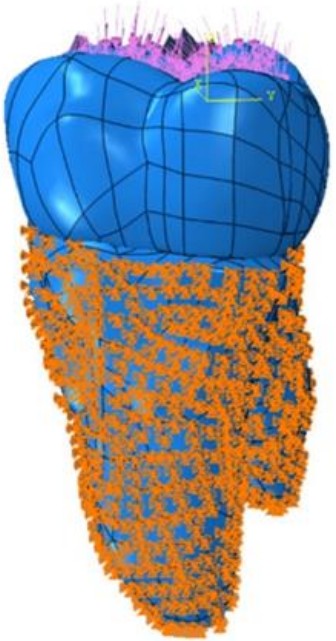

**Figure 4.** Molar tooth model (load and boundary conditions).

An isotropic linear elastic restoration simulation was conducted using the materials. To apply a 600 N load, the pressure to be applied was calculated by measuring the area for each model [28]. FEA was used to assess the stress distribution using Abaqus software (2020 Dassault Systems Simulation Corp, Johnston; RI, USA). In this study, to achieve reliable results, the global size of 0.08 was adopted for the mesh size. The mesh, nodes, and elements used in the FEA for the tooth and cement thickness are presented in Table 3. The analysis was initiated after the geometry and appearance of the mesh were properly meshed and regularized. To achieve the desired number of elements, the main parameter, which was the maximum principal stress, was taken into account. In the subsequent step, the number of elements was doubled, and the effect of this mesh reduction on the mentioned parameter was investigated. This process was repeated until a compromise between time and resources was achieved, without any significant changes in responses with the increase in the number of new networks. At this stage, it was concluded that the solutions converged and that there was no need to use more elements. Increasing the number of elements did not help to enhance the accuracy of the solution but only prolonged the solution process.

**Table 3.** Nodes and elements for tested groups.

| Model | Total Elements | Total Nodes | Mesh Type |
|---|---|---|---|
| 50 μm | 7,428,602 | 1,347,225 | Linear tetrahedral elements of C3D4 |
| 100 μm | 7,445,941 | 1,350,049 | Linear tetrahedral elements of C3D4 |
| 150 μm | 7,457,979 | 1,352,224 | Linear tetrahedral elements of C3D4 |

### 3. Results

The results of the stress accumulation analysis in different layers, including in the enamel, dentin, restoration, and resin cement applied in different thicknesses, are presented in Figure 5. The maximum principal stress (Pmax; MPa) values were utilized for this analysis. The findings reveal that the highest stress accumulation was observed in the enamel layer, followed by the restoration layer, the dentin layer, and the resin cement layer.

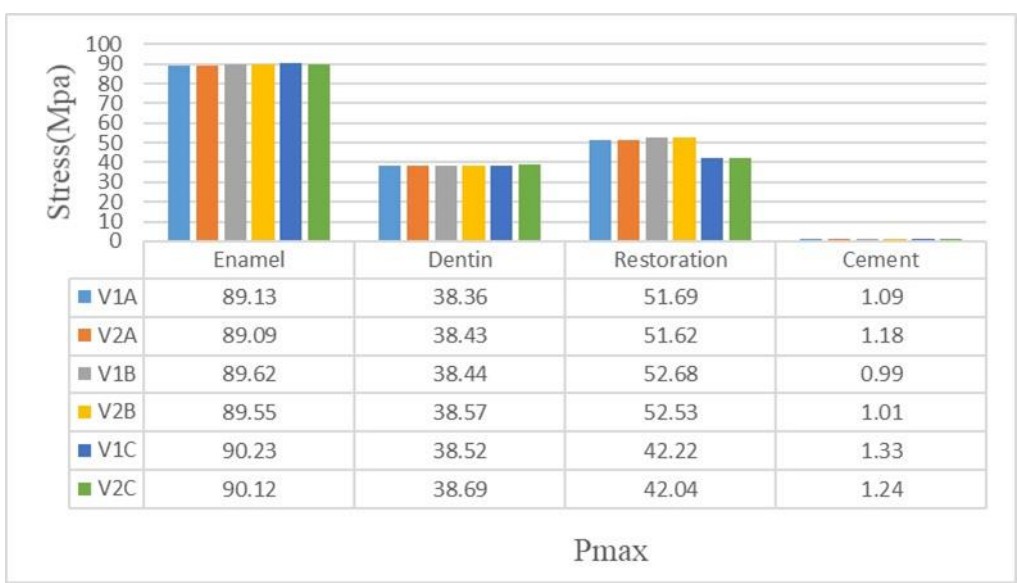

**Figure 5.** Maximum principal stress (MPa) distribution within the enamel, dentin, restoration, and different thicknesses of resin cement in each group under 600 N of total force applied to the occlusal surface.

The results of the stress accumulation analysis are presented in Figure 6, where the maximum principal stress values (Pmax; MPa) are shown for the enamel, dentin, restoration, and the varying thicknesses of the resin cement (V1; Poisson's ratio: 0.35). The resin cement had an elastic modulus of 8.3 GPa and a Poisson's ratio of V1: 0.35. The enamel had the highest stress value (Pmax: 90.23 MPa) when the resin cement with a thickness of 150 μm was used, and the lowest stress accumulation (Pmax: 89.13 MPa) was observed when the resin cement with a thickness of 50 μm was used. In the dentin, the highest stress accumulation (Pmax: 38.52 MPa) was found when the resin cement with a thickness of 150 μm was used, while the lowest stress accumulation (Pmax: 38.36 MPa) was observed when the resin cement with a thickness of 50 μm was used. The restoration had the highest amount of stress accumulation when the resin cement with a thickness of 100 μm was used (Pmax: 52.68 MPa), while the lowest amount of stress accumulation was found when the resin cement with a thickness of 150 μm was used (Pmax: 42.22 MPa). The highest amount of stress accumulation in the resin cement was observed when it was used with a thickness of 150 μm (Pmax: 1.33 MPa), and the lowest amount of stress accumulation was found when the resin cement with a thickness of 100 μm was used (Pmax: 0.99 MPa).

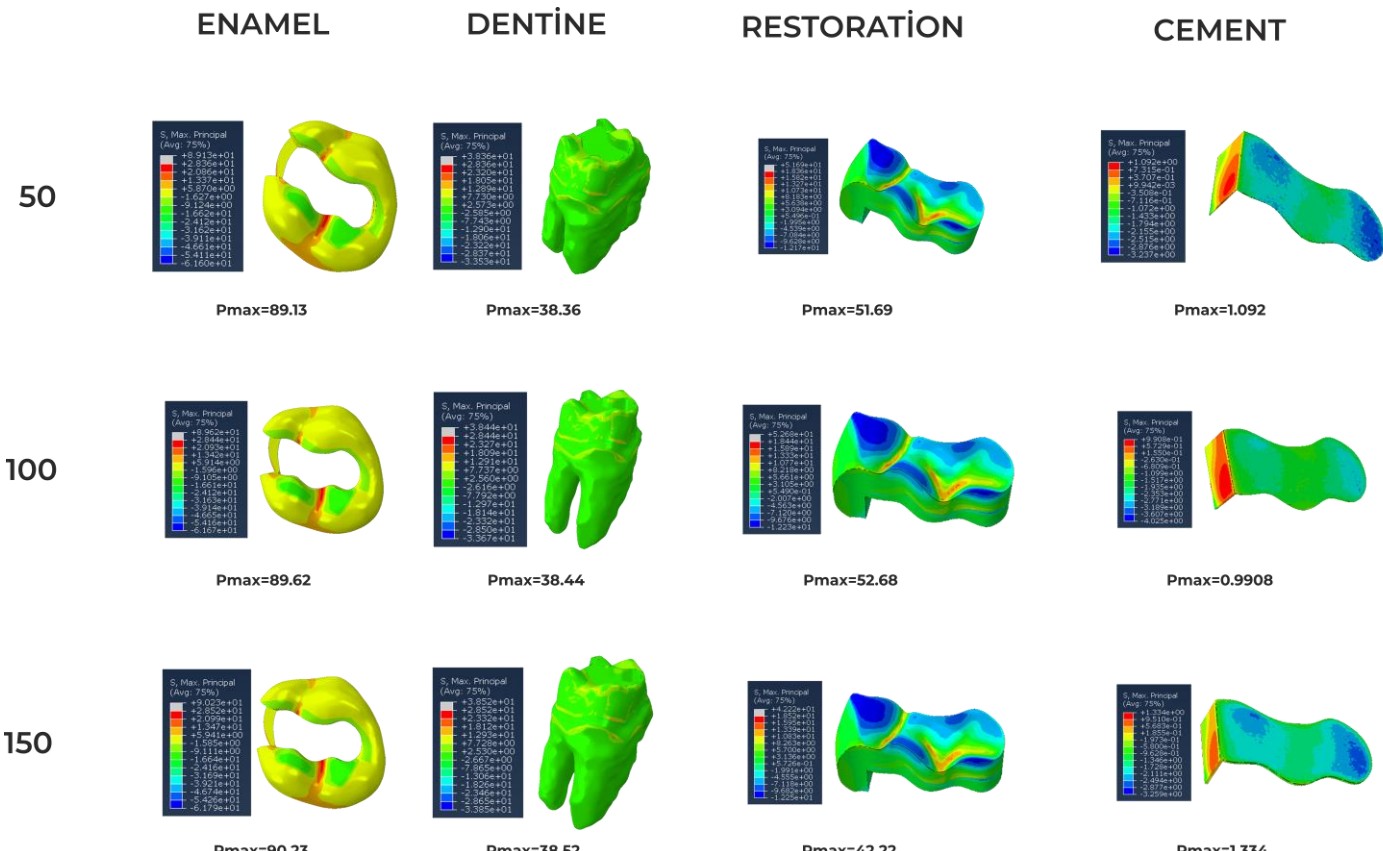

**Figure 6.** Maximum principal stress (MPa) distribution within the enamel, dentin, restoration, and different thicknesses of resin cement under 600 N of total force applied to the occlusal surface (V1; Poisson's ratio: 0.35) with the amalgam.

The maximum principal stress distribution in the enamel, dentin, restoration, and the different thicknesses of the resin cement (V2: Poisson's ratio: 0.24) is shown in Figure 7. In the resin cement with an elastic modulus of 8.3 Gpa and V2, the highest amount of stress in the enamel (Pmax: 90.12 MPa) was found when the resin cement had a thickness of 150 μm. In contrast, the lowest amount of stress (Pmax: 89.09 MPa) occurred when the resin cement had a thickness of 50 μm. In the dentin, the highest amount of stress (Pmax: 38.69 MPa) was found when the resin cement with a thickness of 150 μm was used, while the lowest amount of stress in the dentin (Pmax: 38.43 MPa) was found when the resin cement with a thickness of 50 μm was used.

The highest amount of stress accumulation in the restoration was found when the resin cement with a thickness of 100 μm (Pmax: 52.53 MPa) was used, while the lowest amount of stress accumulation was found when the resin cement with a thickness of 150 μm (Pmax: 42.04 MPa) was used. In the resin cement, the highest amount of stress accumulation was found when the resin cement with a thickness of 150 μm was used (Pmax: 1.24 MPa), and the lowest amount of stress accumulation was found when the resin cement had a thickness of 100 μm (Pmax: 1.01 MPa).

When an elastic modulus of 8.3 GPa with a Poisson's ratio of V1 and an elastic modulus of 8.3 GPa with a Poisson's ratio of V2 were used, the stress on both the enamel and dentin increased with an increase in the cement thickness, while the accumulation of stress in the restoration decreased. The highest stress value for the enamel and dentin was observed at the 150 μm cement thickness for both V1 and V2, whereas the lowest stress value was observed at the 50 μm cement thickness. The amount of stress in the enamel and dentin increased as the cement thickness increased in both resin cements. In both resin cements,

the stress in the restoration increased up to 100 μm cement thickness and then decreased up to 150 μm resin cement thickness, as shown in Figure 5.

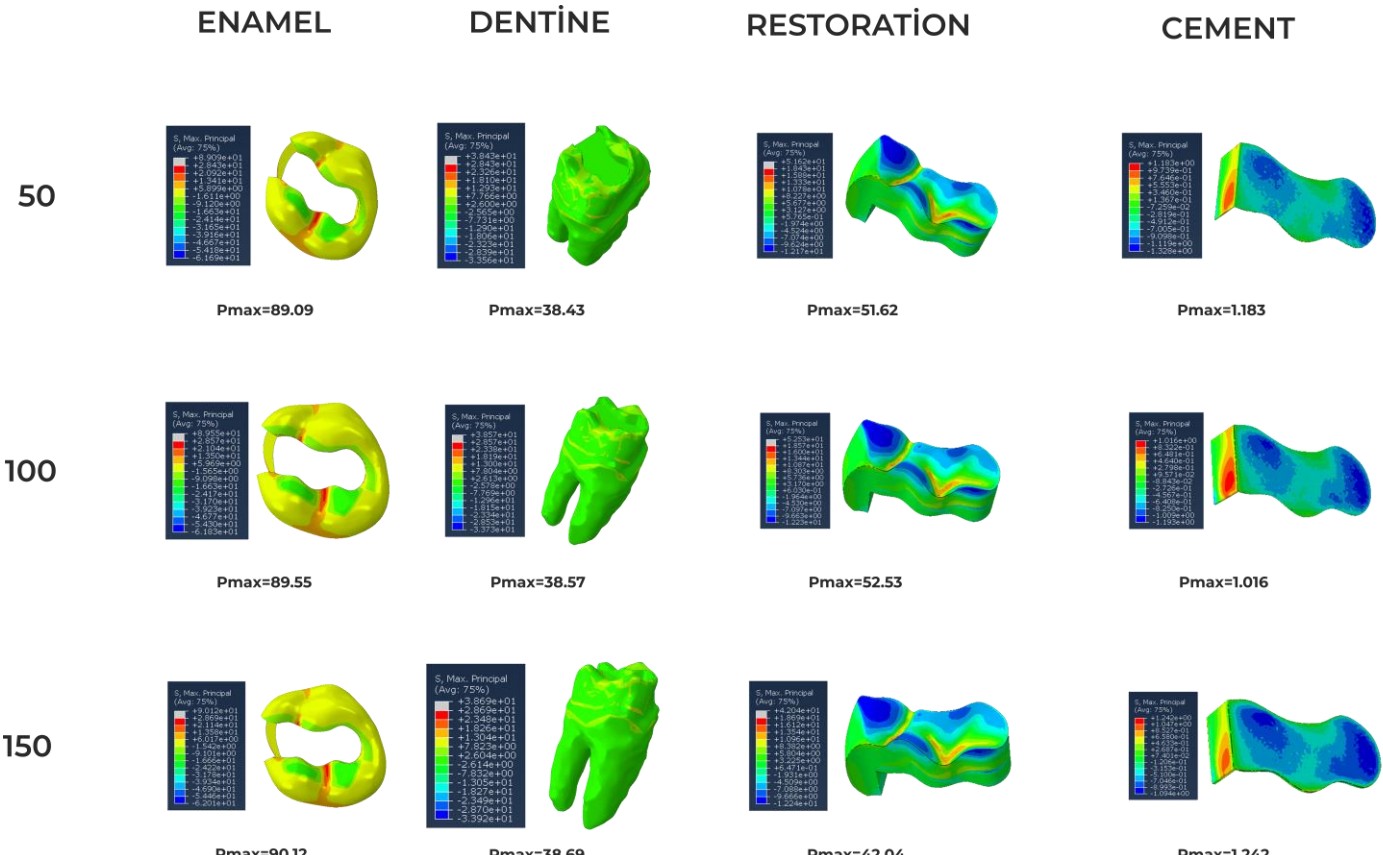

**Figure 7.** Maximum principal stress (MPa) distribution within the enamel, dentin, restoration, and different thicknesses of resin cement under 600 N of total force applied to the occlusal surface (V2; Poisson's ratio: 0.24) with the amalgam.

When the V1 cement was used, the amount of stress in the enamel and the restoration was higher than that when the V2 cement was used, while the amount of stress in the dentin when the V2 cement was used showed higher values than that when the V1 cement was used. Additionally, the stress in the V1 and V2 resin cement decreased up to 100 μm cement thickness and increased up to 150 μm cement thickness. The highest amount of stress for the restoration was observed in both resin cements at the 100 μm cement thickness, whereas the lowest stress value was observed at the 150 μm cement thickness. For both V1 and V2, the highest stress value was observed at the 150 μm cement thickness, whereas the lowest stress value was observed at the 100 μm cement thickness.

## 4. Discussion

This study utilized the finite element stress analysis method to investigate the stress distribution in class II DO cavities filled with dental amalgam and resin cement, with the enamel, dentin, dental amalgam, and resin cement having varying Poisson's ratios and stress values. The results demonstrate that the thickness of the resin cement influenced the concentration of stress at the restoration interface and in the dental tissues. Therefore, the hypothesis was refuted based on the findings of this investigation.

In this study, a 3D dental model was generated using CT-based data. This approach allowed for the creation of an accurate 3D finite element model of a tooth, and it is essential in exploring various clinical issues in dentistry [29,30]. Advancements in milling technology and modeling methodologies have made FEA an effective method in biomechanical

applications. FEA is particularly useful in dentistry when experimental applications do not provide sufficient information [30,31]. FEA provides dependable information regarding stress areas, especially in structures where the analyzed surface is not smooth. It also allows for a swift analysis of intricate computer-designed components and minimizes costs and time by reducing the number of required test subjects [32]. During simulations, our study utilized surface sliding-type contact elements in a 3D FEA. However, different Young's modulus values were considered for the bolus, and a stable occlusive loading force of 600 N was applied [9]. According to a previous study, the average human bite forces in the molar region are 847 Newtons for men and 597 Newtons for women [33]. In general, loads can be applied directly to a tooth/filling through concentrated or distributed means or indirectly using a ball or bite [8]. The quantity, location, and angle of the load are influenced by various factors, such as chewing patterns and food type. Such complex loading conditions can lead to different and potentially more critical stress scenarios. Moreover, an anisotropic elastic material model was utilized to examine all tissue types [34]. In a previous study, it was noted that the finite element analysis method is a valuable tool for researchers and clinicians to enhance and strategize oral health [35]. It is known that molar teeth account for approximately 50% of all dental fractures [36]. As such fractures are common in molars, this study utilized a molar tooth for modeling purposes.

Functional restorations utilizing dental amalgam can have a lifespan of over 60 years if appropriately prepared and maintained, owing to its properties. Therefore, any material aiming to replace it must first be proven mechanically, biologically, practically, and socio-economically superior, which has yet to be achieved [7]. Dental amalgam is one of the most commonly used restorative materials in posterior teeth due to its high fracture resistance and tensile qualities. The chewing forces exerted on dental amalgam are absorbed not only by dental hard tissues but also by periodontal tissues and alveolar bone [8]. Materials with a higher shear modulus and lower compressibility are generally more rigid, but they also tend to be brittle [37]. The elasticity modulus of a material indicates its relative hardness within elastic ranges, and this can be determined through various techniques, such as tensile and compression testing. Restorative materials with a low modulus have been shown to aid in stress release [38]. Restorative materials with a high elasticity modulus can reduce the flexure of tubercles in class II cavities [24]. The final mechanical behavior of class II DO restorations has been studied using a 3D finite element analysis, simulating the effects of different cavity margin angles and occlusal loading conditions with foods of varying stiffnesses [39,40]. This study utilized the finite element analysis method to demonstrate the stress distribution of class II DO amalgam restorations under varying resin thicknesses. However, the polymerization shrinkage of the resin layer was not considered. Maximum principal stress was used as the index to evaluate the materials' stress fracture susceptibility [8,28,31]. Moreover, this method aids in the determination of the maximum tensile stress generated by different loading conditions in various oral tissues, structures, and materials [17,30]. The location of stress can vary significantly due to changes in geometry and the materials used at the interface, while the amount of stress is influenced by the remaining tooth structure and the interface between the tooth and the restoration material [9]. While a healthy tooth is functional, dental tissues are mainly under significant pressure and stress. Only a few parts of the tissues are subject to tensile stress [1], which can cause a concentration of stress in dental material properties [8]. In a previous study, it was stated that enamel absorbs most of the occlusive force and, therefore, shows stresses higher than those absorbed in dentin because enamel is harder than dentin [1]. This study found that enamel had higher localized stresses, whereas dentin had lower and more uniformly distributed stresses. Because the elastic modulus of enamel and materials is not equivalent, stress concentrations result [30]. If the repair material has a higher elasticity modulus, destruction is less noticeable [31]. Enamel, which has the highest adhesive force and elasticity modulus among dental tissues, tends to accumulate more stress while present, and its integrity has a direct impact on restoration longevity [41,42]. A previous study emphasized that the aim of dental treatments should be to restore the function of

enamel and dentin tissues. Enamel has an important mechanical role in the grinding of food. Dentin absorbs the mechanical forces from the enamel, so mechanical properties are important [43]. In our study, the highest stress accumulation was found in enamel tissues.

The cement thickness used in restorations is controversial. A greater adhesive cement thickness can reduce the support from the tooth and cause ceramic breakage, while an adhesive cement thickness with excessively thin layers can adversely affect the longevity of the restoration [44,45]. In previous studies, 70 μm resin cement and 100 μm resin-modified glass ionomer cement [46], 120 μm cement [42], 100 μm resin cement [47], 50 μm cement [48], and 60 μm cements [49] were investigated. Long-term performance in resin cements is said to be optimal with an internal gap size of 50–100 μm [50]. It has been recommended that the interfacial gap size for resin cement should not exceed 100 μm [51,52]. Moreover, an increased film thickness of resin cement can have an impact on the fitting passivity of the restoration, leading to the thickness of the cement thickness being greater than ideal. Considering that the cavity boundaries terminate in dentin, it was believed that the thickness of the resin cement could have an effect on the final outcome. Therefore, we conducted a study using resin cements with thicknesses of 50, 100, and 150 μm. In particular, as the setting process progresses, the Poisson's ratio should decrease significantly as it transforms into a brittle solid [37]. When cement was applied in varying thicknesses and the Poisson's ratio was reduced, the stress accumulation in the enamel and restoration decreased, while the stress in the dentin increased. This indicates that the thickness of the resin cement used may cause differences in stress levels across different structures. When V2A or V2B were used, the resin cement absorbed more stress and transmitted less stress to the enamel and restoration. However, when V1C was used, the resin cement absorbed more stress and transmitted higher stress to the enamel and restoration. In addition, there was less stress accumulation in the restoration when V2 was used. This indicates that both the resin cement and dental tissues have greater stress when using the resin cement at 150 μm thickness.

Increasing the cement thickness led to a higher amount of stress in the enamel and dentin while reducing the stress accumulation in the restoration. It was also found that the stress accumulation in the cement was lower than that in the other groups with a thickness of 100 μm. Resin cement, which has a lower elastic modulus than dentin, increases the stress in dentin to support the material under further distortion [53]. At a resin thickness of 100 μm, the stress accumulation in the dentin was reduced, and the dentin experienced greater stress. However, when the resin thickness increased to 150 μm, the stress in the resin and dentin increased, while the stress in the restoration decreased. The thickness of the cement can lead to variations in stress levels across different structures under the restoration. When using a rigid material, failure is expected to occur in the adhesive interface layer [54]. It is crucial to avoid premature contact and changes that may lead to the premature fatigue of the restoration when using a thinner cement thickness [55]. However, a greater cement thickness may lead to more defects, a poorer micromechanical compatibility, and a higher water absorption than the use of thinner layers. It can also cause a deterioration in the adhesive strength to the substrate [56,57]. However, a thick cement layer can provide a relatively flexible, stress-absorbing layer between the restoration material and the dentin, resulting in low interfacial stress [58,59].

Microleakage remains a concern for class II cavities where the gum edges are in contact with the dentin [17]. A previous study [60] showed that the bonding cement can influence microleakage, with resin cement showing a lower microleakage than other types. Therefore, in this study, resin cement was selected as the resin interface material under the amalgam. We thought that the thickness of the resin cement would affect the result since the cavity boundaries end in the dentin tissue. However, the mechanical properties of the resin cement were considered for simulation within the limitations of the method. Protecting the marginal crest of the first molar tooth is critical in increasing the strength of wrinkle resistance [8]. Furthermore, the oblique crest in the first molar tooth is an essential anatomical structure that plays a crucial role in the chewing process, particularly in resisting

diagonal forces [53]. Our observations revealed an increase in the stress concentration in the axial inner walls of the class II cavity and the oblique crest of the molar tooth. This concentration of stress indicates that it may contribute to the appearance of marginal leakage and secondary caries in the tooth. Additionally, high stresses were detected in the combination of enamel and dentin in the class II cavity [8], raising concerns about possible enamel prism deterioration at this location due to stress.

The excessive stress induced by the restorative material in the enamel may be explained by the differing mechanical properties of the enamel and the material [8,9]. A previous study reported reduced stress values in teeth where materials with elastic modulus values closest to those of enamel were used [24]. In the current study, changing the cements' Poisson's ratios and thicknesses led to the highest stress accumulation in the enamel. However, materials with a high elastic modulus were found to increase the stress in the cement layer, thereby increasing the risk of separation. Additionally, increasing the thickness from 100 μm to 150 μm resulted in significant increases in stresses in both the enamel and dentin, and the lowest stress accumulation was found in the resin cement. A previous study also suggested that material thickness affects the restoration and resin cement biomechanics [57].

In this study, FEA was used to examine the durability and structure of fillings in dental procedures that involve the use of both amalgam and resin cement. However, due to the different properties of both materials, different limitations and approaches may be required for analysis. The dental fillings have limitations such as material modeling, geometric limitations, load limitations, and surface properties. However, when done correctly, FEA can provide important information about the durability, stress distribution, and fracture behavior of amalgam fillings. This was an in vitro study and may not completely replicate the complex oral environment. The inclusion of chewing forces and boundary conditions in an actual tooth model could have led to different findings. Simplified restoration and tissue characteristics were used, ignoring important anatomical features, such as enamel rods and dentin tubules. The study did not investigate stress distributions in the periodontal ligament or bone. The simplified restored system was only modeled for parameter study purposes, and, therefore, it may not fully represent clinical situations. Additionally, the physical shrinkage of dental restorations, as well as thermal stressors, was not considered. These limitations should be considered when interpreting the findings of this study.

## 5. Conclusions

The stress experienced by dental tissues and restorative materials is influenced by the thickness of the resin cement applied under the amalgam restoration. An increase in a cement's Poisson's ratio resulted in less stress accumulation in the cement (50–100 μm), but this may increase stress accumulation in the restorative material and dental tissues. Therefore, the thickness of the cement plays a crucial role in minimizing the stress accumulation in dental tissues.

This study found that stress accumulation was more pronounced in the edge of the restoration, as well as in the oblique crest and distal axial inner wall in the amalgam class II disto-occlusal restoration. It is recommended that an overly thin cement thickness should not be used under restorative materials. Additionally, using resin cement with low Poisson's ratio values in clinical practice may reduce stress in dental tissues and restorations. Clinicians should be aware that areas with a high stress concentration can lead to marginal leakage and tooth fractures. In dental biomechanics, FEA is a dependable method for exploring dental materials.

**Author Contributions:** Conceptualization, H.Y.G. and M.F.; methodology, H.Y.G., M.F. and R.M.; software, R.M.; validation, R.M.; formal analysis, R.M.; investigation, H.Y.G., M.F. and R.M.; resources, M.F.; data curation, H.Y.G., M.F. and R.M.; writing—original draft preparation, Y.D.F., M.F. and S.K.; writing—review and editing, Y.D.F., M.F. and S.K.; visualization, R.M.; supervision, H.Y.G.; project administration, H.Y.G. All authors have read and agreed to the published version of the manuscript.

**Funding:** This research received no external funding.

**Institutional Review Board Statement:** Not applicable.

**Informed Consent Statement:** Not applicable.

**Data Availability Statement:** The data presented in this study are available upon request from the authors.

**Conflicts of Interest:** The authors declare no conflict of interest.

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
