# Peer review of "Comparison of Resin Cement’s Different Thicknesses and Poisson’s Ratios on the Stress Distribution of Class II Amalgam Restoration Using Finite Element Analysis"

_applsci, doi:10.3390/app13074125_

Round 1

Reviewer 1 Report

Dear editor,

The author of the paper which title is “Comparison of Resin Cements Different Thicknesses and Pois sons Ratios on the Stress Distribution of Class II Amalgam Restoration Using Finite Element Analysis” introduced  to using three-dimensional finite element analysis to evaluate the effect of type of cements different thicknesses and Poissonsrations on stress distribution at enamal, dentin, restoration, and resin cement of computer-aided designed class II dis-to-occlusal cavity. This work may be suitable to the scope of Applied Sciences However, there are some problems in this paper, mainly including the problem of traditional research methods and lack of innovation. Author should consider the following comments before publishing the paper.

1. In the first paragraph of the introduction, what was the purpose of the authors to mentioned the ideal dental material in dental caries?

2. The authors used a Class II cavity generated by a computer model without forming a linear relationship between Poisson's ratio and stress of the resin cement by three-dimensional finite element analysis.

3. In order to more clearly describe the stress distribution at the tooth interface during occlusal loading, the authors should calculate different values of external applied pressure to evaluate the stress and strain distribution of the tooth structure and resin cement.

4. The Poisson's ratio value in Figure 7 is inconsistent with the corresponding value marked in the paper.

5. The author should further analyze and explain the nodes and elements of Table 3, test group.

6. To demonstrate the superiority of the author's work, the results of three-dimensional finite element analysis should be compared with similar literature of numerical analysis in bioengineering and dentistry.

7. Results and discussion parts are inadequate according to citation and analysis in detail. These sentences should be supported by the literature studies.

8. The format of reference is standardized and the literature of the last two years is quoted.

In the conclusion, in addition to summarizing the actions taken and results, please strengthen the expianation of their significance.

Author Response

Dear reviewer(s),

We are deeply grateful to the anonymous reviewers for your constructive and insightful comments to the our study. According to your feedbacks, we have revised the manuscript again. We have reviewed these comments respectively and after that revised them in detail. The paper revised and edited the spelling of words and trouble some sentences by a translate service. 

Thanks in advance for your interest.

Best regards,

According to reviewers’ comments, our revisions are respectively as follows in detail:

Reviewer(s):

We are sincerely thank to the reviewer for adress to critical points in strengthening our manuscript. We have summarized all revisions below (by showing color in the manuscript):

Reviewer 1

R1-1. In the first paragraph of the introduction, what was the purpose of the authors to mentioned the ideal dental material in dental caries?

We revised the first paragraph of the introduction. We removed the mentioned the ideal dental material in dental caries. We added new sentences and references.

R1-2. The authors used a Class II cavity generated by a computer model without forming a linear relationship between Poisson's ratio and stress of the resin cement by three-dimensional finite element analysis.

R1-3. In order to more clearly describe the stress distribution at the tooth interface during occlusal loading, the authors should calculate different values of external applied pressure to evaluate the stress and strain distribution of the tooth structure and resin cement.

In order to provide 600N loading, the pressure to be applied was calculated by measuring the area for each model. This load value was quoted from the literature (De Abreu RAM, Pereira MD, Furtado F, Prado GPR, Mestriner W, Ferreira LM. Masticatory efficiency and bite force in individuals with normal occlusion. Arch Oral Biol. 2014;59(10):1065-74). (The 600 N load was calculated as F=P/a by calculating the area of the chewing area and applied as pressure in the finite element analysis)

R1-4. The Poisson's ratio value in Figure 7 is inconsistent with the corresponding value marked in the paper.

We revised that sentence.

R1-5. The author should further analyze and explain the nodes and elements of Table 3, test group.

We added new sentences Methods section. (You can look at the paragraphs in highligted color).  

Three different types of models were used to create test groups.

‘Different combinations were simulated, including the same modulus of elasticity for two cement groups (V1 and V2; Table 1) and different thicknesses (50 µm (A), 100 µm (B), and 150 µm (C)). As a result, six study groups were created (Table 2).’

R1-6. To demonstrate the superiority of the author's work, the results of three-dimensional finite element analysis should be compared with similar literature of numerical analysis in bioengineering and dentistry.

We added more similar literatüre about tested-selected conditions discussion parts. (You can look at the paragraphs in highligted color).  

R1-7. Results and discussion parts are inadequate according to citation and analysis in detail. These sentences should be supported by the literature studies.

We added more detail about tested-selected conditions methods and discussion parts . (You can look at the paragraphs in highligted color).  

R1-8. The format of reference is standardized and the literature of the last two years is quoted.

The literature was expanded and supported by the literature studies. (about tested-selected conditions methods and results)

R1-9. In the conclusion, in addition to summarizing the actions taken and results, please strengthen the expianation of their significance.

We revised and rephrased that conlusion part.

Reviewer 2 Report

The manuscript describes the study applying the finite element stress analysis method to determine the amount of stress on enamel, dentin, restoration, and resin cement in the amalgam class II disto-occlusal (DO) cavity using resin cement with different Poisson’s ratios and thicknesses.

The topic is relatively novel. However, the scope is narrow. It is not clear how stress concentration is related to the cause in the emergence of secondary caries in the tooth. What’s more, due to the missing instrument operating parameters in Section 2, the reproducibility is low.

Extensive English editing is necessary.

Thus a major revision is recommended before further consideration in Applied Sciences.

Some suggestions:

1. This theoretical work predicting Young’s modulus and Poisson’s ratio of dental amalgam should be cited, doi: j.intermet.2009.12.004

The work discussing the influence of physical-mechanical properties of luting cements in general on the restoration stress distribution should also be cited 10.1038/ncomms9631

2. Why were periodontal ligament and the bone (maxillar) not considered? As they also respond to the stress exerted, they should be considered.

3. L129-137 is repetitive.

4. “The amounts of stress that come to enamel….” Come is not the right word to use here.

5. Table 3 and Fig. 5 the ordering of groups is not consistent.

6. English usage is poor, e.g. L43-47, L61, L85, L124, there are more.

7. What are the Poisson’s ratios and thicknesses of common cement used for this application? Why did you choose these in the analyses? 0.35;V1, 0.24;V2 is confusing, should be V1: 0.35; V2: 0.24.

Author Response

Dear reviewer(s),

We are deeply grateful to the anonymous reviewers for your constructive and insightful comments to the our study. According to your feedbacks, we have revised the manuscript again. We have reviewed these comments respectively and after that revised them in detail. The paper revised and edited the spelling of words and trouble some sentences by a translate service. 

Thanks in advance for your interest.

Best regards,

According to reviewers’ comments, our revisions are respectively as follows in detail:

Reviewer(s):

We are sincerely thank to the reviewer for adress to critical points in strengthening our manuscript. We have summarized all revisions below (by showing color in the manuscript):

R2-1. This theoretical work predicting Young’s modulus and Poisson’s ratio of dental amalgam should be cited, doi: j.intermet.2009.12.004

The work discussing the influence of physical-mechanical properties of luting cements in general on the restoration stress distribution should also be cited 10.1038/ncomms9631

Thanks for your contribution. The influence of physical-mechanical properties of luting cements was quoted from the literature (discussion part) A citation was made from the section concerning resin cements.

Young’s modulus and Poisson’s ratio of dental amalgam was cited relevant citation and was added Table 1.

R2-2. Why were periodontal ligament and the bone (maxillar) not considered? As they also respond to the stress exerted, they should be considered.

Previous studies (Ref 8 and Ref 27)  was not considered. Therefore this study did not investigate stress distributions in the periodontal ligament and bone.

Ref 8 and Ref 27 were added methods section.

‘As the periodontal ligament (PDL) was not modeled, fixed and pinned boundary conditions were utilized to simulate roots that are fixed in the bone [8][27]. A single tooth and tooth type were used without simulating the periodontal ligament or bone. The mechanical boundary conditions (symmetry/antisymmetry/encostre) were selected using the "create boundary condition" tab in the load part of the Abaqus program. The effect of the periodontal ligament was ignored, and the tooth was pinned (U1=U2=U3=0) from the enamel-cementum junction to the apical region (Fig. 4).’

R2-3. L129-137 is repetitive.

We revised and removed repetitive sentences.

R2-4. “The amounts of stress that come to enamel….” Come is not the right word to use here.

The manuscript revised and edited the spelling of words and trouble some sentences by a translate service. 

R2-5. Table 3 and Fig. 5 the ordering of groups is not consistent. Maximum main stress distribution in resin cemented restorations of different thicknesses.

Three different types of models were used to create test groups.

‘Different combinations were simulated, including the same modulus of elasticity for two cement groups (V1 and V2; Table 1) and different thicknesses (50 µm (A), 100 µm (B), and 150 µm (C)). As a result, six study groups were created (Table 2).’

R2-6. English usage is poor, e.g. L43-47, L61, L85, L124, there are more.

The manuscript revised and edited the spelling of words and trouble some sentences by a translate service. 

R2-7. What are the Poisson’s ratios and thicknesses of common cement used for this application? Why did you choose these in the analyses? 0.35;V1, 0.24;V2 is confusing, should be V1: 0.35; V2: 0.24.

We added new referances and sentences in discussion part. The use of resin cements under dental amalgam can reduce tubercle fractures and affect the stress accumulation in dental tissues. Therefore, the thicknesses indicated on the basis of literature studies were used;

‘The cement thickness used in restorations is controversial. Thicker adhesive cement thickness can reduce the support from the tooth and cause ceramic breakage, while excessively thin layers of adhesive cement thickness can adversely affect the longevity of the restoration [44][45]. Previous studies, 70 μm resin cement and 100 μm resin modified glass ionomer cement; [46] 120 μm cement; [42] 100 μm resin cement; [47] 50 μm cement; [48] and 60 μm cements [49]  were investigated. Long-term performance in resin cements is said to be optimal with an internal gap size of 50-100 μm [50]. It has been recommended that the interfacial gap size for resin cement should not exceed 100 μm [51][52]. Moreover, an increased film thickness of the resin cement can have an impact on the fitting passivity of the restoration, leading to cement thickness conditions that are much greater than ideal. Considering that the cavity boundaries terminate in dentin, it was believed that the thickness of the resin cement could have an effect on the final outcome. Therefore, we conducted a study using resin cements with thicknesses of 50, 100, and 150 μm.’

Round 2

Reviewer 1 Report

The authors of the paper which title is “Comparison of Resin Cement’s Different Thicknesses and Pois son’s Ratios on the Stress Distribution of Class II Amalgam Restoration Using Finite Element Analysis”has responded point as per my proposed comments, agreed to accept and publish in the Applied Sciences. However, the following issues remain.

1. The font line spacing in Table 1 and Table 2 is inconsistent.

2. The author should unify the drawing format.

3. The section of Results and discussion parts is inadequate, which needs more reliable data.

Author Response

Dear reviewer,

According to reviewers’ comments, our revisions are respectively as follows in detail:

Reviewer(s):

We are sincerely thank to the reviewer for adress to critical points in strengthening our manuscript. We have summarized all revisions below (by showing color in the manuscript):

R1-1.  The font line spacing in Table 1 and Table 2 is inconsistent.

We revised the font line spacing in Table 1 and Table 2.

R1-2. The author should unify the drawing format.

We arranged the drawing format.

R1-3. The section of Results and discussion parts is inadequate, which needs more reliable data.

In this study, current developments in the field of dental FEA analysis were discussed. Unfortunately, due to the limited number of recent studies in this area, the discussion was also limited. Based on the findings, new sentences were added to emphasize certain points in the discussion.

‘When V2A or V2B are used, the resin cement absorbs more stress and transmits less stress to the enamel and restoration. However, when V1C is used, the resin cement absorbs more stress and transmits higher stress to the enamel and restoration. In addition, there is less stress accumulation in the restoration when V2 is used. This indicates that both the resin cement and dental tissues have greater stress deposition for the resin cement at 150 μm thickness.’

Limitation part:

‘In this study, FEA can be used to examine the durability and structure of fillings in dental procedures that involve the use of both amalgam and resin cement. However, due to the different properties of both materials, different limitations and approaches may be required for analysis. The dental fillings has limitations such as material modeling, geometric limitations, load limitations, and surface properties. However, when done correctly, FEA can provide important information about the durability, stress distribution, and fracture behavior of amalgam fillings.’   

Reviewer 2 Report

The majority of the issues have been addressed.

What is the FEA mesh size?

Author Response

Dear reviewer,

According to comments, our revisions are respectively as follows in detail:

Reviewer(s):

We are sincerely thank to the reviewer for adress to critical points in strengthening our manuscript. We have summarized all revisions below (by showing color in the manuscript):

R2-1. What is the FEA mesh size?

We added the FEA mesh size in method part. ‘In this study, to achieve reliable results, the global size of 0.08 was adopted for the mesh size.’
